# SAR Deep Learning Sea Ice Retrieval Trained with Airborne Laser Scanner Measurements from the MOSAiC Expedition

Karl Kortum[1, 2], Suman Singha[3, 4], Gunnar Spreen[2], Nils Hutter[5], Arttu Jutila[6, 5], and Christian Haas[5]

[1]Remote Sensing Technology Institute, German Aerospace Center (DLR), Bremen, Germany
[2]Institute of Environmental Physics, University of Bremen, Bremen, Germany
[3]National Center for Climate Research, Danish Meteorological Institute (DMI), Copenhagen, Denmark
[4]Department of Geography, University of Calgary, Calgary, Canada
[5]Alfred-Wegener-Institut (AWI), Bremerhaven, Germany
[6]Finnish Meteorological Institute (FMI), Helsinki, Finland

**Correspondence:** karl.kortum@dlr.de

**Abstract.** Automated sea ice charting from Synthetic Aperture Radar (SAR) has been researched for more than a decade and still, we are not close to unlocking the full potential of automated solutions in terms of resolution and accuracy. The central complications arise from ground truth data not being readily available in the polar regions. In this paper, we build a dataset from 20 near coincident X-Band SAR acquisitions and as many Airborne Laser Scanner (ALS) measurements from the Multidisci-plinary drifting Observatory for the Study of Arctic Climate (MOSAiC), between October and May. This dataset is then used to assess the accuracy and robustness of five machine learning based approaches, by deriving classes from the freeboard, surface roughness (standard deviation at $0.5m$ correlation length) and reflectance. It is shown that there is only a weak correlation of the radar backscatter and the sea ice topography. Accuracies between $44\%$ and $66\%$ percent and robustnesses between $71\%$ and $83\%$ give a realistic insight into the performance of modern convolutional neural network architectures across a range of ice conditions over 8 months. It also marks the first time algorithms are trained entirely with labels from coincident measurements, allowing for a probabilistic class retrieval. The results show that segmentation models able to learn from the class distribution perform significantly better than pixel-wise classification approaches by nearly $20\%$ accuracy on average.

## 1 Introduction

Sea ice classification from remote sensing and especially SAR instruments have been used for monitoring the Arctic sea ice for multiple decades, with automation being proposed as early as the mid eighties by Fily and Rothrock (1986). However, even with the inception of advanced machine learning methods and modern data analysis, there does not yet exist a universally reliable classifier to retrieve sea ice classes from radar imagery. The potential for such a classifier is obvious: Humans are not able to match the speed and precision of an automated algorithm. Until now, however, this potential has yet to be fully unlocked; Human-generated ice charts (for an overview regard the World Meteorologial Organizations overview by JCOMM (2017)) are still dominant in operational usage, despite the considerable amount of research that has been focused on the subject. These products unfortunately can provide only coarse approximate labels of the sea ice. For cross-cutting research, a more detailed and

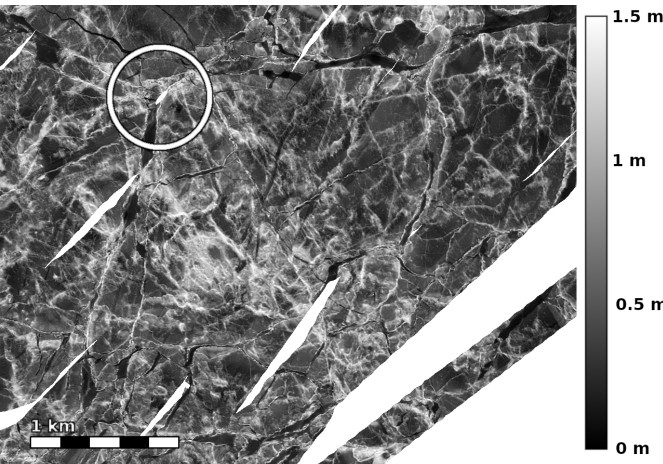

**Figure 1.** Section of ALS measured freeboard over the MOSAiC floe on April 8, 2020. RV Polarstern can be seen in the center of the white circle. Brighter values correspond to higher freeboard values whereas white areas indicate no data. The displayed freeboard range is 0 to 1.5 metres.

higher resolution of classification would be preferable and should be possible given the spatial resolution of the SAR sensors. As a human analyst generating an ice chart has only limited time to annotate a SAR scene, such high-resolution labels are not contained in the ice charts. Leads, for example, hot spots of ocean and atmosphere interaction and thus of particular interest for

the energy budgets, are generally not labelled in operational ice charting. At this point, with many different classifiers having been proposed and developed including domain knowledge based and centre-pixel as well as semantic segmentation models (e.g. Kwok et al. (1992); Soh and Tsatsoulis (1999); Hara et al. (1995); Karvonen (2004); Ressel et al. (2015); Doulgeris (2015); Johansson et al. (2020); Lohse et al. (2021)) one must ask the question why no meaningful direction has yet established itself in the ongoing research. The answer to the question - aside from the complexity of the subject - is twofold. Firstly and most

important is the state of the data. Although we have a great wealth of satellite SAR acquisitions of the sea ice in diverse states and conditions, we lack the corresponding ground truth information. Secondly, the constantly varying and difficult-to-predict drift and deformation of sea ice makes it nearly impossible to image the same area of sea ice over longer time series to evaluate any proposed classifiers' robustness. The latter is particularly true for high-resolution imagery. These two shortcomings open the development of classifiers performing at or near the resolution of the SAR imagery to a plethora of different challenges

because we have almost no way to test, iterate and improve retrieval algorithms in a structured manner. This stifles the rate at which progress in the field can be made or even recognised.

     On a mission to fill gaps in our knowledge about the Arctic sea ice and its climatology, the MOSAiC expedition launched in the autumn of 2019 and the ship Polarstern spent a year adrift with the ice pack. Aboard, interdisciplinary teams of scientists worked to collect as many data as possible, which will help to further our understanding of one of our earth's most remote

regions. With the mission came the unique opportunity to collect exactly the type of ground truth over a long time period, that

is needed to test sea ice retrieval algorithms, with satellite-borne SAR data being acquired at the same time. An overview of the snow and ice related activities is given in Nicolaus et al. (2022).

Ice and snow transects from Itkin et al. (2021) or drilling hold the most detailed information of the underlying ice. Unfortunately, the spatial extents of these measurements are too sparse to be used for comparison with the satellite acquisitions. Aerial measurements taken from helicopters, such as the Airborne Laser Scanner (ALS) data products by Hutter et al. (2022, 2023) being used in this approach (Fig. 1) provide information about the height of the snow and/or ice surface above the local sea level, i.e. freeboard, and surface reflectance at scales of kilometres to tens of kilometres. These data are therefore a prime candidate to extract ground truth information for ice classification based on roughness and thickness. Because of the efforts made during the MOSAiC expedition and the subsequent collocation for this work, the dataset used here is far larger than any previously used data derived from measurements. However, it still suffers from a loss of generality from being constrained to certain region. It is none-the-less most likely the most complete (as in large) collocated dataset that we will be able to synthesise at least until another expedition of the scope of MOSAiC comes along (which might not even happen before the first ice free summer in the Arctic).

One prominent emerging method of segmenting image data are machine learning based approaches based on convolutional neural networks, such as published in Simonyan and Zisserman (2015); He et al. (2015); Liu et al. (2022); Ronneberger et al. (2015); Zhou et al. (2018, 2019). Advancements in the field of machine vision are being made at a rapid pace, able to leverage the improvements in chip design and the increasing amount of data that are being generated. The image-like properties of SAR acquisitions mean that this knowledge is transferable to the ice classification domain (e.g. Boulze et al. (2020); Ullah et al. (2021); Wang and Li (2021); Kortum et al. (2021, 2022)). Historically, this has been done with texture extraction and subsequent dense neural networks as in Ressel et al. (2016); Singha et al. (2018); Murashkin et al. (2018), pixel-wise classification using image classifiers based on convolutional neural networks as by Boulze et al. (2020); Ullah et al. (2021) and segmentation models that are able to segment an entire patch simultaneously as detailed in Wang and Li (2021). Previous studies have also sought to use passive microwave data to derive labels Radhakrishnan et al. (2021), which consequently concentrate on much larger scales.

In this study, we will use the unique opportunity provided by 20 instances of near-coincident (7 hours time difference on average) ALS and SAR data over a period of 8 months to compare a variety of machine learning-based classification approaches in terms of classification accuracy and robustness on classes delineated directly from measurements. For the first time, we have accurate, high resolution sea ice topography measurements of freeboard and surface reflectance with high spatial overlap and low time differences between acquisitions to truly test the capability of retrieving freeboard and (above snow) surface roughness based sea ice classes from SAR data. In contrast to existing ALS and SAR datasets, such as produced in Singha et al. (2018), the MOSAiC experiment provides the opportunity to monitor the same ice across a large temporal time span at high resolution. The amount of colocations achieved here is significantly greater than in previous studies, which enables the training of deep learning models requiring large datasets. Concretely, the questions we are trying to answer are: How do different CNN architectures perform on labels derived from measurements (not human interpretation) and how does that influence future algorithm choices if the aim is to produce classifications near the resolution and detail of the SAR measurements?

## 2 Methodology

### 2.1 The Data

The SAR component of the analysis is made up of TerraSAR-X X-band acquisitions in StripMap (SM) mode. The intensity scenes are normalised to $\sigma_0$ and calibration is performed as per the product specifications in Fritz et al. (2007). The resulting scenes have a pixel spacing of 3.5 metres and a native radiometric resolution of 16 bit. Both HH and VV bands are acquired by the satellite simultaneously. This configuration of polarisations has been shown to yield valuable information for ice classification in Ressel et al. (2016)Geldsetzer and Yackel (2009). As only 2 bands can be acquired simultaneously, the cross-pol band is not present in the data. Each combination of two channels will have some shortcomings, however, so this needs to be accepted. The footprint of a single scenes is typically around 50x15km. Other SAR data is not available at the resolution and frequency to enable high spatial overlap with ALS measurements (in terms of number of pixels) at small enough time differences. At higher wavelengths, especially L-Band we would expect at a higher correlation between radar backscatter and ALS derived surface roughness measured at spatial intervals of 0.5 metres. This would probably translate to higher classification accuracy for deformed ice. In terms of generality of the derived results, the complexity of the spatial distribution of classes and hence the core results derived in this study, we would expect to translate to coarser resolutions and other frequencies.

The ALS data from Hutter et al. (2023, 2022) from 20 scenes (appendix A) between October 2019 and May 2020 are used to delineate sea ice classes. The data were acquired by flying a mow-the-lawn pattern over the ice near the MOSAiC central observatory. The resulting ALS grid has a geospatial resolution of 0.5 metres. For midwinter flights in high latitudes of >85°N, the post-processing of the helicopter INS/GPS data failed and ALS data processing was performed using a lower frequency real-time navigation solution with metre-scale undulations in GPS altitude that propagated to the surface elevation retrieved from the ALS. The undulations in the computed freeboard could be minimised using a correction calculated from swath-to-swath overlap. It should be noted that the local standard deviation of the freeboard is left intact by these processing artefacts and can still be used to derive a parametrisation of the local surface roughness, where these undulations are present. An additional measurement aside from freeboard is the surface reflectance at the wavelength of the laser (1064 nm), which is useful to identify regions of young ice that have not yet been covered by snow. For the acquisitions with unphysical undulations in the freeboard measurement, freeboard was not used to delineate class labels. Instead, only classes which could be inferred from the surface roughness and reflectivity were used. The footprint of a single flight is typically around 5x10km.

*Colocation*: For each ALS grid, the first step for co-locating with SAR data is to find the SAR acquisition that is closest to the ALS measurement time, whilst still having substantial spatial overlap. Then, by using the Polarstern ship to determine a common coordinate system, the two measurements are fused by assigning each ALS data point to the closest SAR pixel (see. Kortum et al. (2021); Hendricks (2019).) In the common coordinate system, this means that the two measurements are in the same TerraSAR-X grid cell relative to the ship. Because of the difference in resolutions (0.5m ALS and 3.5m SAR), we obtain approximately 49 points of ALS measurements per SAR pixel. The freeboard and roughness are then computed as the respective mean and standard deviation of these points. Investigation showed that the median and mean of the local distribu-

tions where on average within less than a percent of the span of the distribution. This lends confidence, that the distribution is roughly symmetrical and thus the mean and standard deviation describe the statistical nature adequately. Using the Polarstern as an origin of the common coordinate system is sensible, as we have accurate GPS positioning and heading to account for ice drift and rotation. The matching of the two products using this method was accurate to a couple of metres. To further improve the accuracy of colocation, a final translation and rotation was then determined manually. Afterwards, the features overlapped

perfectly at (TerraSAR-X) pixel resolution. The accuracy of co-location is made possible by more than daily TerraSAR-X SAR acquisitions of the MOSAiC floe, which helps keep the time differences between satellite and helicopter measurements small.

*Determining labels*: We have categorised the measured sea ice into three classes. A label is given for each SAR pixel, for which ALS information is available. For ease of reference, we are giving them names which are easier to contextualise.

However, the exact definitions of the classes is given here. They are fully given by the ALS measurement. The three classes are: Open water and young ice (OW/YI), level first-year ice (LFYI) and deformed first-year and multiyear ice (DFYI/MYI). These classes we define as follows (see Fig. 2 for a visual aid):

- OW/YI: Ice whose reflectance (range corrected target echo amplitude) is significantly lower than that of the surrounding snow covered ice. Typically values around -7dB were used as a threshold value and adjusted manually if needed. Note,

that finer separation here is not possible from the data alone, but from reports of scientists on the expedition we know that most ice in this class will have already formed a thin ice layer and entirely open water was very rare during the flights.

- LFYI: Snow-covered ice with a surface roughness (standard deviation of freeboard measurements at scales of the ALS grid (0.5m) calculated over one SAR pixel ($3.5 \times 3.5$ m$^2$)) of less than 1 centimetre or a freeboard value lower than the

higher inflection point in the freeboard distribution (typically around 40 centimetres).

- DFYI/MYI: Snow covered ice with a surface roughness of more than 10 centimetres or a freeboard greater than the higher inflection point in the freeboard distribution.

As detailed above, ice types are identified by thresholds in the reflectance, surface roughness or freeboard. The thresholds for

the roughness and freeboard are indicated in the histograms in Fig. 2 by the different background colors. We can infer the probabilities of lying above or below a threshold for every pixel by assuming a Gaussian distribution of ALS freeboard and reflection measurements at each SAR pixel. From the 49 ALS measurements mapped to one SAR pixel, we compute the mean and standard deviation of the freeboard and can then compute the probabilities of lying below or above the globally defined freeboard thresholds by using the error function. Explicitly, we integrate the area under the curve of the estimated Gaussian

probability density function, above and below the threshold. An example is shown in Fig. 3. Thus we obtain 'soft labels' which give the probabilities of belonging to a certain class, rather than discrete classes. Assuming a Gaussian distribution allows us to also infer uncertainties of the surface roughness. One could have classified each of the 49 ALS measurements mapped to

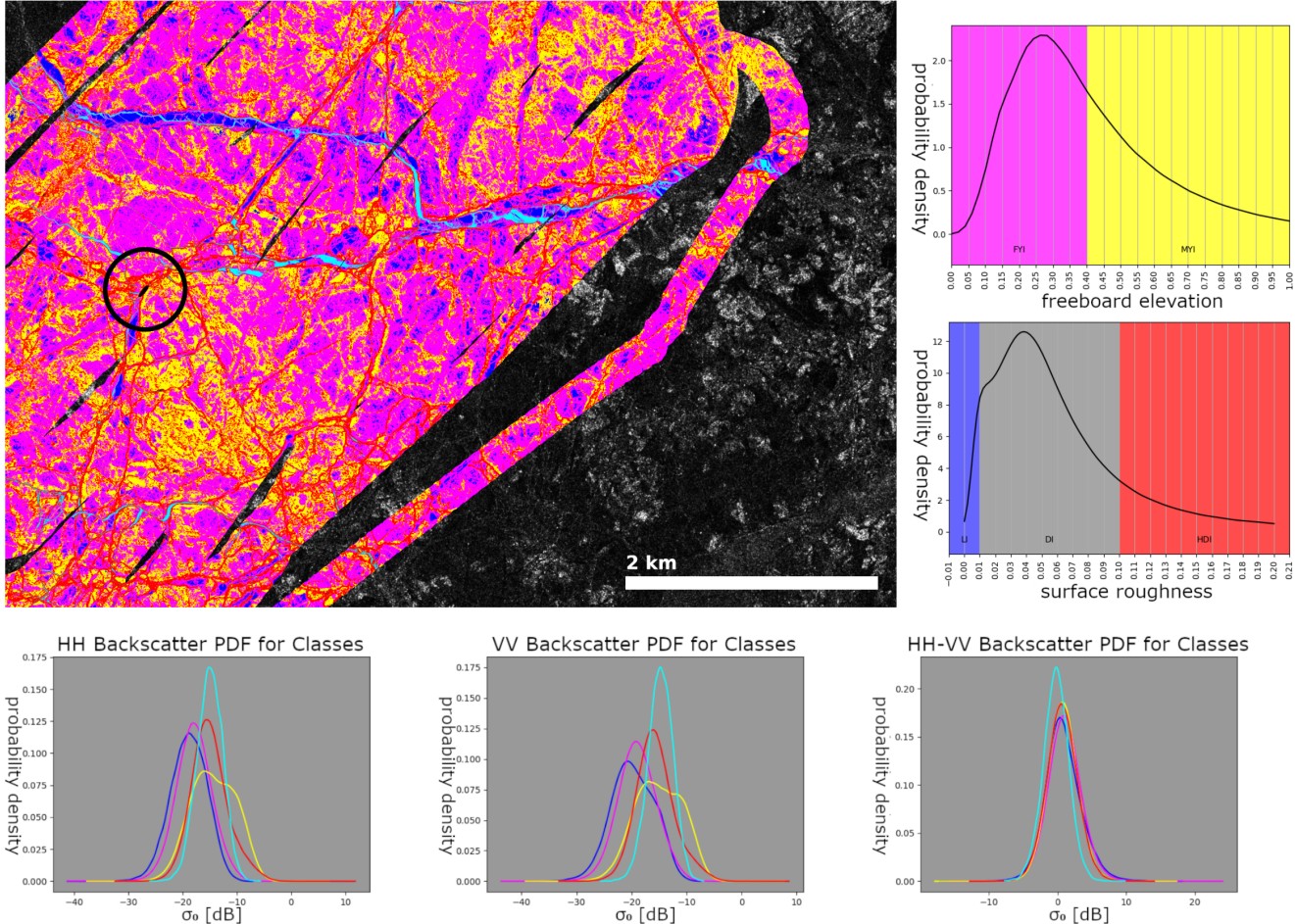

**Figure 2.** Derived labels from the ALS acquisition on the April 8, 2020 overlayed on the HH channel of the near-coincident SAR measurement (left) and estimated probability density functions from the distributions of freeboard and surface roughness (in this case this is the local standard deviation of the freeboard) (right). *Yellow* indicates ice with a higher freeboard than the high inflection point of the distribution. *Magenta* is ice with a lower free board than that. *Red* are areas with higher surface roughness than 10 cm. *Blue* areas are ice with surface roughness of less than 1 cm. *Cyan* areas have reflectivity indicating no snow cover (less than -7dB Echo Amplitude). For this study, yellow and red, as well as magenta and blue classes are combined. The grey background of the surface roughness distribution denotes the region that was not used to identify ice classes, as there was considerable mixing in this parameter region. At the bottom approximate probability density functions (PDFs) for the sigma nought backscatter of each class across the different polarization configurations are shown. Note that no two classes can be reliably separated using backscatter alone. All of the pdfs have been smoothed with Gaussian kernel smoothing.

one SAR pixel and then used the relative occurrences as probabilities. However, this simplification to a Gaussian distribution leads to an inaccuracy of the probabilities (derived from freeboard) of only $\approx 0.16\%$ on average, but significantly increased computational efficiency.

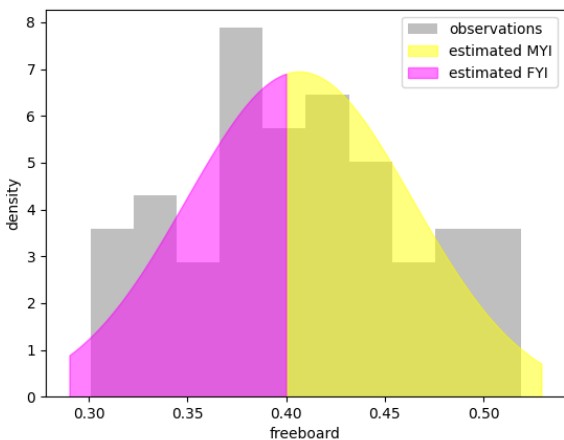

**Figure 3.** Soft labels are derived for one SAR pixel by assuming a Gaussian distribution (colored) of the 49 ALS observations (grey histogram) inside of it and then integrating the area under the pdf curve above and below the threshold. In the given example the probabilities are close to 50%.

The derived labels from each scene are split into three mutually exclusive connected subsets. By connected we mean, that in all but edge cases pixels are neighbouring ones from the same subset. The training set is made up of 80% of labels whilst the test and validation sets consists of 10% each. The validation data is used only to decide when to stop training. All subsets (test, training, validation) contain data from every scene. Imbalances of the classes were handled by balancing the dataset for pixel-wise classifiers and weighting the classes inversely to their frequency for the segmentation approaches, where an entire patch is segmented at once. Thus the training of the networks is set up so that performing equally well for each class yields the lowest loss. As the classes are not balanced in the labels, better performance would certainly be achieved on the training data set without balancing the classes, but it would hinder the generalisation of the classifier and make the results more difficult to interpret. As generalisation to a larger space of ice conditions is a property we would like to be reflected in the results as directly as possible, balancing was undertaken here.

In Figure 4 the correlation between backscatter and surface topography measurements is shown. It becomes evident immediately, that the backscatter characteristics are only very weakly correlated with the topography and thus separation using the backscatter alone would surely be futile. This is further underlined by looking at the backscatter distributions of the delineated classes from the flight on April 8th (fig. 2, bottom), where the correlations are relatively average in regards to all other flights. Here it is again obvious that the backscatter characteristics are only somewhat valuable for class separation. Thus the information needed to classify accurately must in large part be derived from contextual data.

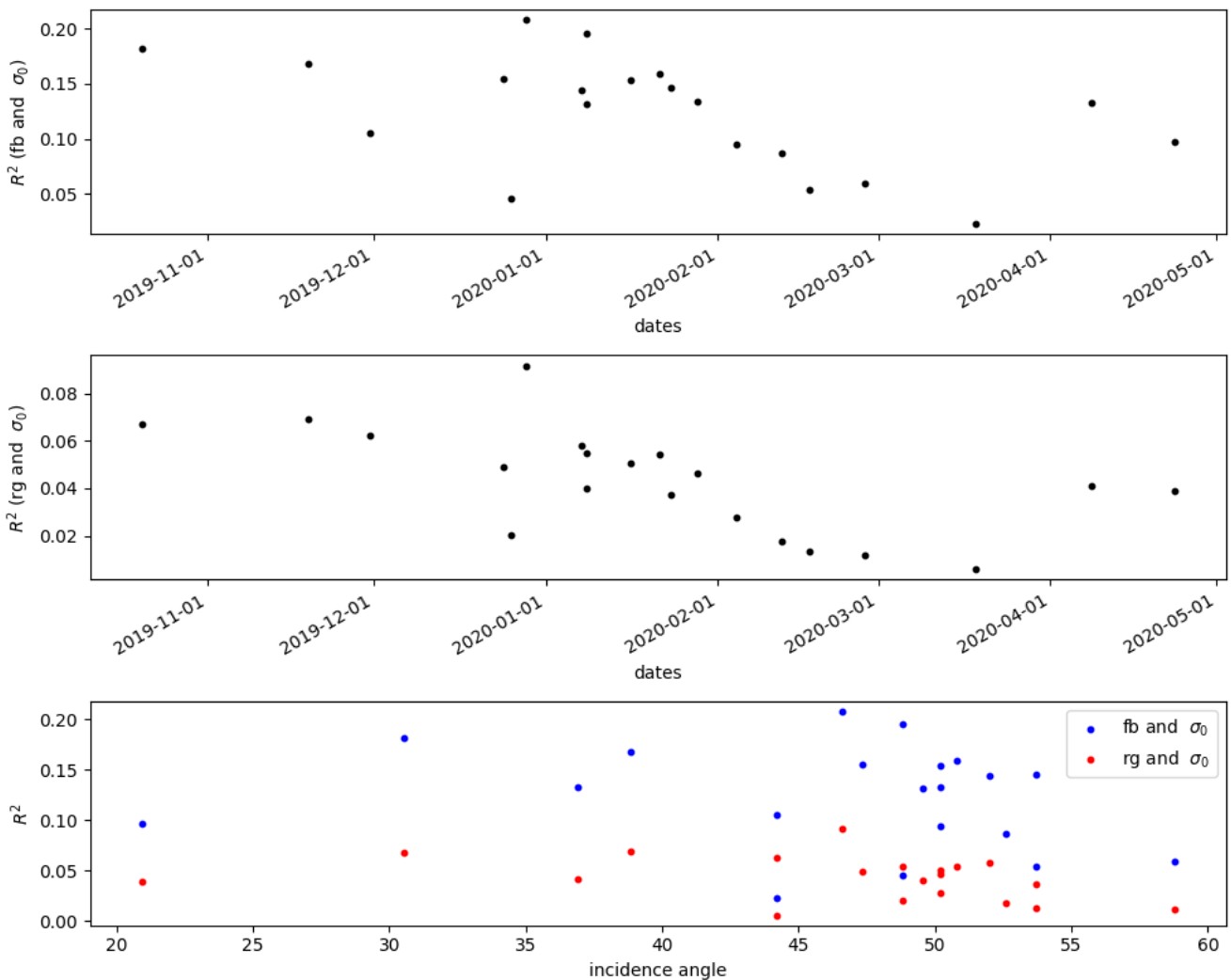

**Figure 4.** Evolution of correlations between freeboard [fb] or surface roughness [rg] and HH SAR backscatter [$\sigma_0$]over time. Bottom plot shows correllations plotted against incidence angle. Note that the surface roughness is measured at the snow atmosphere interface and at correlation lenghts of 0.5 metres, whilst the SAR sensor is most sensitive to the ice snow interface and roughness at correlation lengths at the wavelength of the sensor, which is only 3.1 centimetres. The same analysis with VV channels gives very similar results.

## 2.2 Robustness

To test the robustness of each classifier, we will follow the same steps outlined in Kortum et al. (2022). In brief: Using the Polarstern as an origin, a 3km x 3km region around the ship is used as the robustness test set. This area has been identified in 162 TerraSAR-X (TSX) SM scenes from different days. The robustness is then defined as the probability of each pixel being classified the same as in the previous and subsequent acquisitions (time between acquisitions is typically one day). Taking into account that the surface conditions are changing over time and that Polarstern was not perfectly stationary, this approximation of the robustness will serve only as a lower bound of the actual robustness of the classifier. In summary we are operating under the assumption that in a time period of two days, the percentage of ice that has changed class (e.g. through deformation) is significantly smaller than the percentage of ice that has remained in the same class. Note that this test is only sensible for the two solid ice classes and not for the OW/YI class, which is too dynamic on a daily timescale to be analysed in this manner. The robustness is first computed for the two classes and their average is used as an indicator for the network's robustness.

## 2.3 The Network Architectures

In this paper, we will compare five different architectures: two established image classifiers in the VGG16 developed by Simonyan and Zisserman (2015) (ice classification in e.g. Khaleghian et al. (2021b)) and the ConvNext network proposed by Liu et al. (2022) (an improvement over ResNet, used for SAR sea ice classification in e.g. Song et al. (2021)), a custom CNN (cCNN) pixel-wise classifier by Kortum et al. (2022) specifically designed for ice classification and two established segmentation models in the Unet by Ronneberger et al. (2015) (SAR sea ice classification in e.g. Nagi et al. (2021); Ren et al. (2022)) and Unet++ proposed in Zhou et al. (2018, 2019) (used in e.g. Murashkin and Frost (2021)). These first three (VGG16, ConvNext, cCNN) and last two (Unet, Unet++) models have one fundamental difference: Classification approaches (VGG16 etc.) are given a patch and are then asked to predict the class of the centre of the image. Segmentation approaches (Unet etc.) are tasked to produce a label for every pixel in the patch at the same time. The exact specifications of all the models can be found in the appendix. A short overview over the core features of the models is given here.

The oldest of the models, the VGG16 was originally developed for image classification. It uses convolutions of filter size three and maxpooling layers to reduce the spatial dimensions. Finally the model architecture is completed with fully connected layers. The core idea is to extract increasingly complex features in the convolutional blocks and only to keep the most prominent once in the maxpool operations. Finally this generates a feature vector of complex spatial features, which is used by the fully connected layers to infer a class.

The ConvNext model is an implementation of some core advantages that self-attention based transformer models have brought to the image classification domain in a convolutional framework. The model uses skip connections which were made a staple in large networks by the ResNet architecture and convolutional blocks using large filters, an inverted bottleneck and depth-wise convolutions inspired by transformer models. It is a more modern design achieving significantly higher scores on image classification tasks, yet is not designed for centre-pixel classification as is typical in sea ice retrieval applications.

The custom CNN model uses multiple zoom levels as inputs and is constrained with few number of parameters and fewer layers than the other model. The central design philosophy here is to avoid overfitting through model constraint. This model architecture was optimised with performance on coarser human annotations in mind.

The Unet model follows a similar approach to the VGG16 in the initial (encoding) stage of the network, where features are disseminated by convolutional layers. However, in the second part of the model these features are upsamled again to create a two dimensional semantic segmentation map. The core advantage over centre-pixel classifiers lies in the fact that inter-label dependencies and relations can be learned and exploited by the architecture.

The Unet++ keeps the encoding-decoding framework of the original Unet but adds more intermediate layers at various scales and fuses the features from multiple scales to make a more informed prediction. We chose to average over the features of the multiple output layers in the deep supervision part of the model.

It is important to note that as little as possible was changed about the architectures themselves to keep true to the models that were proposed in the original publications - otherwise the results would be harder to interpret with the efforts and successes of optimisation techniques being an unknown factor.

## 2.4 Training

During training, the networks are tasked with minimising the categorical crossentropy between the output and the label distributions. This allows us to fit the probabilities of each class occurring at each pixel, which we can infer from the ALS measurements. Minimising the cross-entropy gives the same result as minimising the Kullback-Leibler Divergence (KLD). As this serves as a benchmark and comparison of these models concerning their applicability for sea ice retrieval, no further optimisations have taken place. For each of the model architectures, ten separate instances are from scratch. Training is stopped using a spatially independent validation set (10% of data), after the model hasn't improved since the last 100,000 training samples. Testing is done on another spatially non-overlapping $10\%$ of data, the remaining (disjoint) $80\%$ of data are used for training (see figure 5) for details. Training multiple instances allows some additional insight into the variabilities. The ingested SAR data are pre-processed by converting each band to sigma nought and then applying a logarithm. The incidence angle is provided in a third channel. The adam optimizer proposed by Kingma and Ba (2017) is used to train the models, with the learning rate set to $10^{-4}$. The size of each patch to be classified is chosen to be 256x256 pixels, except for the cCNN which receives input patches at various scales (a 5x5, a 16x16 and a 64x64 pixel patch).

## 3 Results

The performance of different network architectures can be seen in table 1. They paint a clear picture of segmentation models' (Unet, Unet++) improvement over centre-pixel classification approaches. Of the pixel-wise classification approaches, the custom CNN classifier performed best on unseen test data, yet it was still significantly inferior to the segmentation models. We speculate that part of the reason for this is the high spatial resolution of the labels, as we get a label for every pixel from most of the ALS measurements. The pixel-wise classifiers cannot make use of any relationships between or spatial properties

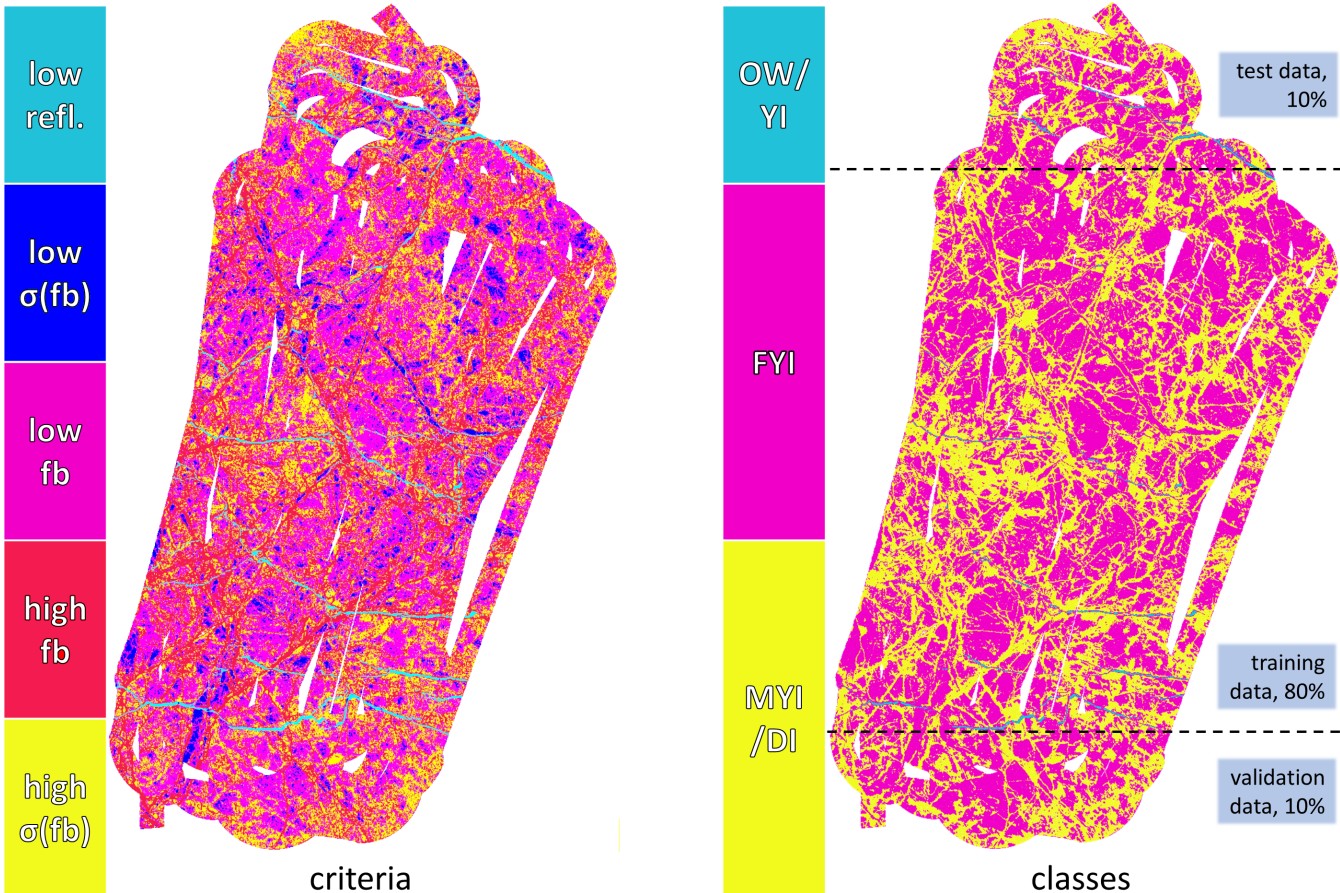

**Figure 5.** Split of training, test and validation data demonstrated on a helicopter product from the 23rd of April. The left hand side shows the different criteria (as in figure 2) and the right hand side shows the derived training data with splits indicated by the dotted lines. The same procedure was carried out for all 20 flights.

of labels, like shape, sparsity and correlations. This seems to be detrimental to their performance. Except for the cCNN which was designed to avoid overfitting, the other centre-pixel classifiers show a large discrepancy between training and test scores, whilst the semantic segmentations models have generalised much more effectively.

A more detailed analysis of the output of different models (Fig. 6) shows, how the VGG16 and ConvNext models struggle to relate all the information of the patch to only the classification of the central pixel, leading to a diffuse-looking classified scene. This seems most pronounced for the ConvNext model. A possible reason for this are the larger convolutional kernels (7x7 in contrast to 3x3) used in the architecture. (A retraining with a kernel size of 3x3 confirmed this increased the average accuracy by around 5%). The cCNN seems to struggle with using contextual data to separate rough ice and young ice. In general, the predicted probabilities at each pixel are higher in the non-dominant class, leading to a seemingly different colour palette in this

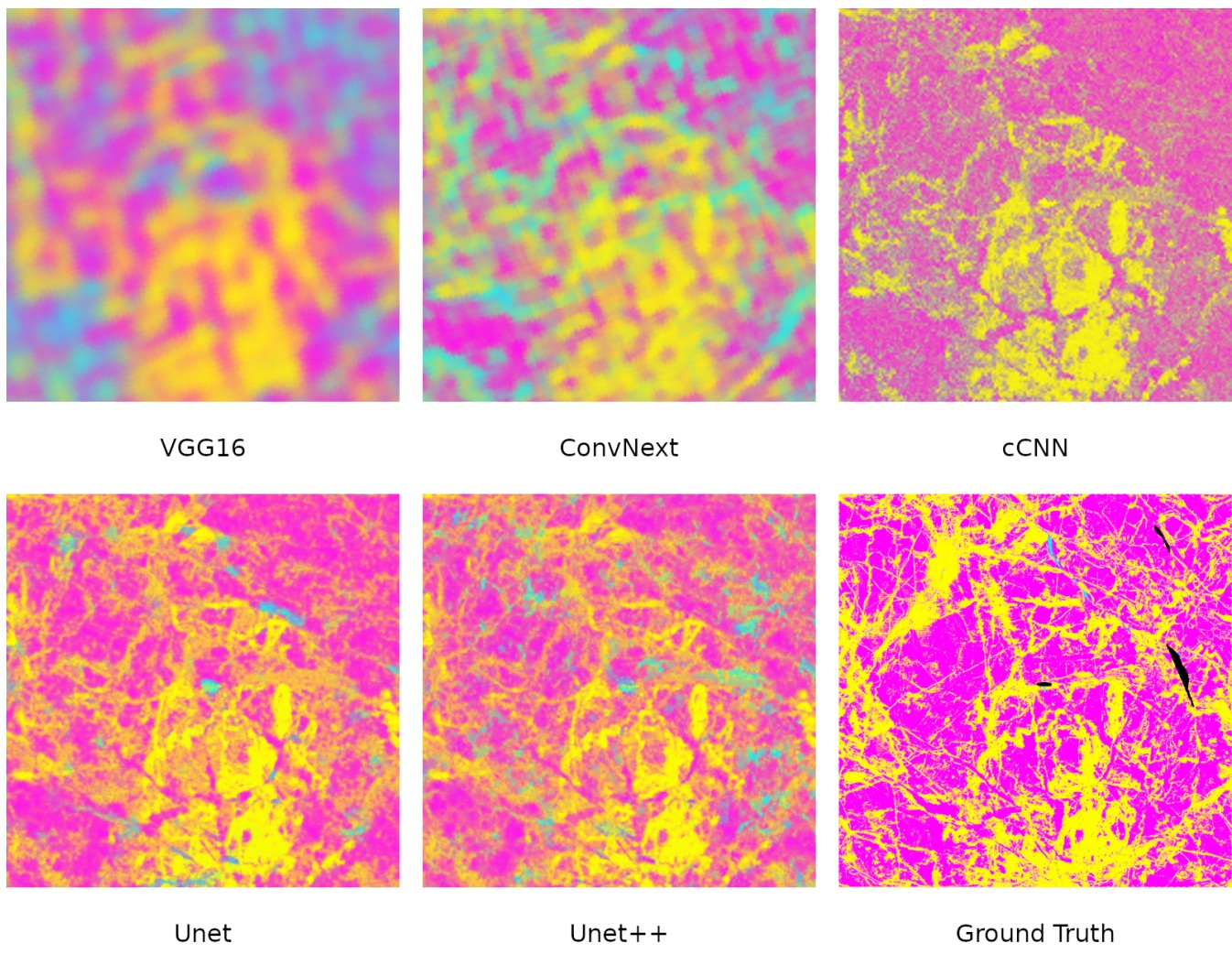

**Figure 6.** Comparison of classifications from different models, randomly selected from the ten instances trained. Colours are the same as the classes discussed above, but the intensity is given by the predicted probabilities, so mixed colours can occur. This can be seen most easily in the cCNN classification (c). The scene was acquired over the Polarstern (center of the images) on January $14^{th}$. The false colour composition consists of HH, VV and HH/VV channels, normalised with a $tanh$ function. The area shown is a 6 by 6 kilometre square.

| Model | train acc. [%] | test acc [%] | train KLD | test KLD | rb. [%] |
|-------|---------------|--------------|-----------|----------|---------|
| VGG16 | 57.80 ± 7.07 | 44.00 ± 1.79 | 0.6468 ± 0.0873 | 0.838 ± 0.0416 | 80.31 ± 5.59 |
| ConvNext | 56.63 ± 5.21 | 44.72 ± 1.47 | 0.704 ± 0.0742 | 0.8784 ± 0.0368 | 81.45 ± 5.6 |
| cCNN | 47.14 ± 1.92 | 47.65 ± 1.85 | 0.7635 ± 0.0188 | 0.7724 ± 0.0269 | 71.12 ± 15.85 |
| Unet | 67.84 ± 2.39 | 65.18 ± 1.08 | 0.5769 ± 0.0307 | **0.6057 ± 0.0316** | **83.12 ± 1.96** |
| Unet++ | 66.06 ± 1.6 | **66.22 ± 1.3** | 0.6104 ± 0.0201 | 0.6374 ± 0.022 | 82.82 ± 1.33 |

**Table 1.** Network performances on the independent test set after training. For brevity, we shortened accuracy to 'acc' and robustness to 'rb'. The means and standard deviations are computed from the 10 models in the population for every architecture. Best-in-category results on independent test sets are highlighted in bold font. Ten instances were trained for every model. The Unet and Unet++ architectures show significantly better performance than the other models tested.

visualisation. The Unet and Unet++ classifications are largely similar. Some difficulty in the separation of deformed and young ice signatures persists as can be seen in the mixing of yellow and cyan areas.

It is also worth pointing out that the very same cCNN and a VGG16 performed at accuracies around 85-95% on manual labels in Kortum et al. (2022), illustrating the difference between training and testing on quantitatively measured labels in contrast to human-generated annotations. In Ren et al. (2022), the Unet is reported to perform sea ice and open water separation on manual labels at 93-95 % accuracy. Wang and Li (2021) report accuracies of 96 % for the same task, using ice charts as training data and test data and 94% accuraciy when comparing to an operational sea ice cover product (Interactive Multisensor Snow and Ice Mapping System, IMS). Murashkin et al. (2018) show classification accuracies of the Unet++ around 96% on manuallly labelled training and test data across 6 classes

Whilst the mean KLD's are in accordance with the accuracies, the spread (std) of the KLD's across the model populations seems to be very similar across all models and there is no clear gap between segmentation and classification approaches. Overall, we cannot say that one model converges more reliably than another - as would be suggested by the accuracies alone. It is also apparent, that the cCNN does not perform well in the robustness scores on this dataset. This model is considerably smaller than the others (in terms of parameter count) and was heavily optimised using a different dataset, which seems to have come at a cost of flexibilty/generality of the architecture. The spread of robustnesses of the segmentation models seems to be considerably smaller that those of the generative models - additionally indicating these approaches are more reliable for ice classification from SAR.

The classifications (e.g. in Fig. 7) show a very plausible set of results, that align with the observations of members on board the expedition. The fine labels at high resolution seem to have transcended into a similarly detailed classification map. The examples in Fig 8 also illustrate a general increase of deformation in the first year ice: The magenta FYI area close to Polarstern, marked by a black square in figure 7, is getting progressively more deformed as time progresses (detail in Fig. 8). The areas most prone to error seem to be the OW/YI classifications. This is to be expected as they are naturally the most sparse in the training data set. Additionally, they are very dynamic, which leads to extremely diverse backscatter properties that can be exhibited, in turn making them more difficult to classify.

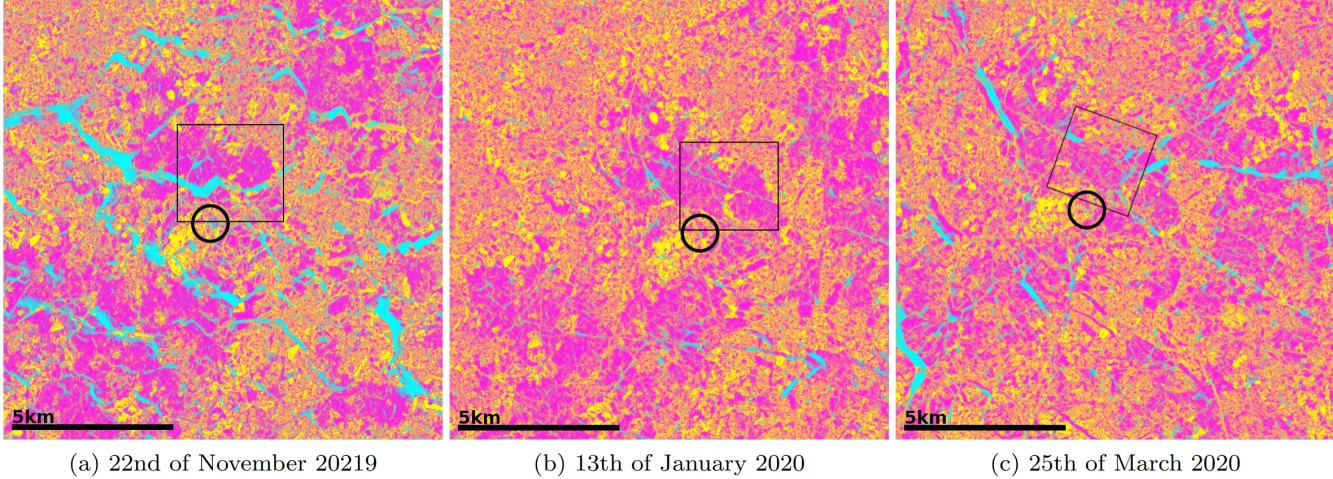

| (a) 22nd of November 20219 | (b) 13th of January 2020 | (c) 25th of March 2020 |

**Figure 7.** Collection of classified subscenes (Unet, pixel spacing = 3.5m) including the MOSAiC floe, after a storm (a), in calm conditions with some shearing indications (b) and with some breakup of the ice cover visible (c). The Polarstern location is indicated by the black circle. The DFYI/MYI class probability is displayed in yellow, the LFYI probability in magenta and the OW/YI probability is cyan. The black square marks the area shown at full resolution below (Fig. 8)

We also observe decreasing correlation of backscatter and surface topography variables from the onset of the expedition until early April - particularly during January and February (4), where the MOSAiC expedition was met by numerous storms. Some of the decorrelation can be accounted for because of snow accumulation and redistribution, but it is difficult to quantify this phenomenon. However, the incidence angle of the scenes also changes and the increase of incidence angle beyond 45 degrees is shown to lead to a continuous decrease in correlation between ice surface characteristics and SAR backscatter.

It should also be noted that the use of one-hot encoded labels leads to a decrease of $8\%$ accuracy in comparison to smooth/probabilistic labels for the Unet architecture.

## 4  Discussion

The top models in our investigation perform at around 68% accuracy on the test data set (Tab. 1). The segmentation models predictions are approximately 20% more accurate than the classification models. The only concrete difference between these models is that the segmentation approaches can learn from the distribution of labels, which appears to be highly important to generalise to unknown regions. The centre-pixel classifiers show a much larger difference between test and training sets. Even the highest accuracies measured here are considerably lower than what many author's report for algorithms trained with human-made labels. To understand these discrepancies, we will discuss the main differences between these measured labels and human annotation.

The measured labels used in this study have some underlying difficulties, because we do not know the snow depth and density, we do not know how strong the correlation of freeboard and ice-thickness is and cannot eliminate this error. Also the

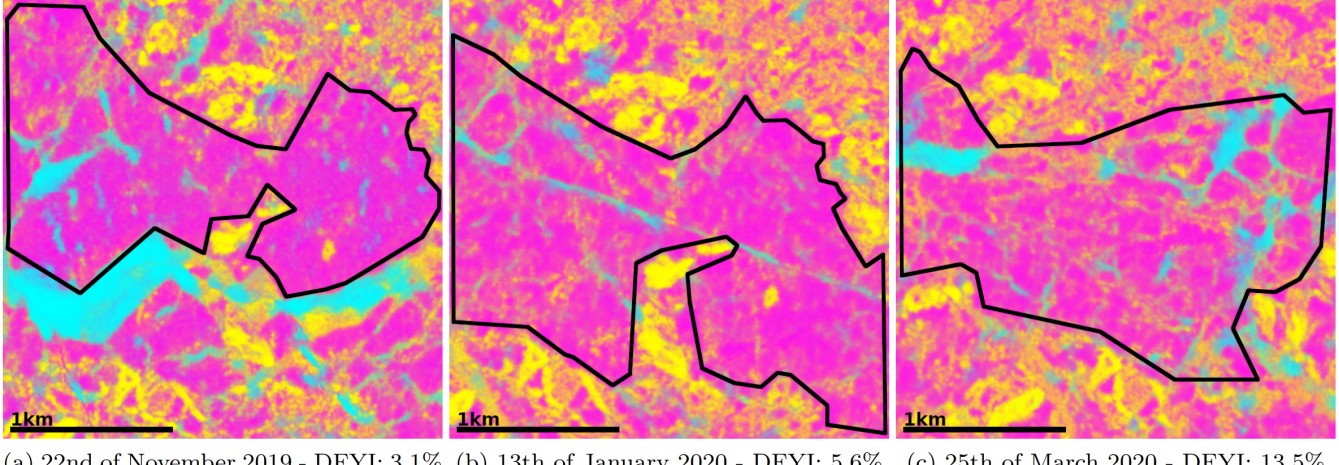

(a) 22nd of November 2019 - DFYI: 3.1%  (b) 13th of January 2020 - DFYI: 5.6%  (c) 25th of March 2020 - DFYI: 13.5%

**Figure 8.** Full resolution excerpt from the scenes show in Fig. 7. The classified images reflect the increased deformation of the first year ice area over time accurately, as the DFYI occurence rises. The DFYI fraction is computed inside of the black border. In the first scene some misclassification of the open lead (cyan) as older, deformed ice (yellow) is seen (outside of the area we are computing the DFYI fraction in) - this is a common issue in SAR sea ice retrieval as the backscatter can become very similar, as also reported by e.g. Guo et al. (2023).

reflectance used to disseminate young ice and open water is based purely on the coverage of the surface being snow free and thus not directly correlated with ice age: if thin ice has formed the atmospheric conditions will dictate whether or not snow has gathered on top or if the bare ice is visible to the sensor. Thus the quality of labels could still be improved on, if more information were available.

To assess the impact of the individual thresholds (e.g. the location of the inflection point in the freeboard distribution) we also evaluated the top-performing Unet architecture on the same dataset, but excluding points near the thresholds. To do this we did not consider labels, where the certainty of the most probable class was lower than 90%. For example regions with local standard deviation of approximately 3cm, that means that points within 6cm of the thresholds are not considered and the exact value of the thresholds have little bearing on the data considered. In case of the test dataset, these data points account for 24.1% of all data. Under these circumstances, the average accuracy of the unet model is 72.5% which is an increase of only 4.18% although 24.1% of the least certain labels where removed. Thus we can conclude that the exact location of the thresholds had only marginal impact on model performance, lending increased confidence that the model performances are representative of performance evaluated against ground truth.

When comparing the data presented here with human annotations/ice charts, one must mention the resolution. In our case, every individual pixel gets its own class and there is no semantic grouping of pixels into the same class based on proximity or likeness. This is a stark contrast to ice charts, where the labels are made up of only few polygons per scene. Even when not training from such ice charts, humans generating training data for algorithms at high resolution generally limit themselves to areas which they can confidently identify. Not much can be said about the correctness of these labels per se, but one should

keep in mind that in these instances, the accuracy achieved by the classifier is constrained to those easy-to-identify regions and are therefore not representative of the classifier's performance on the whole. Because of the size of SAR acqusitions obtaining labels at pixel resolution from human annotation is not feasible. The great advantage that labels from measurements have is that they are truly indicative of performance on the entire scope of ice conditions in the scene (every pixel is labelled, thus there is no selection bias). Only by holding the testing of our high resolution retrieval algorithms to this standard can we show with certainty when an improved method of classification is developed, but of course to do so we are lacking available data sets.

This study had only a small effective study region and a large temporal span to test the diverse conditions. Overall the constancy of the ice in the scenes should only improve the classifiers' performances. Unfortunately the 20 helicopter flights are not quite enough to make meaningful statements about temporal changes in performance, as the differences in performance will be outweighed by the local conditions in the scene. Additionally seasons in the data where one would expect the classification to be most difficult (freeze-up and pre early melt onset) are only very sparsely represented in the data. This means the contributions of the data sparsity, seasonality and spatial variability cannot be meaningfully separated.

In the summer season the ice surface is dominated by wet snow, bare ice and melt ponds and more open water is found between floes. The spatial distribution of classes is very distinctive between the surface types, so one can expect the main result of the difference between centre-pixel classifiers and segmentation models to persist.

In most data-driven approaches to classification, the performance of the classifier is limited by the quality of the labels. Therefore, one should be careful when using manually labelled data, such as ice charts, as ground truth. These practices are common in the current research - as not many other sources of labels are available. However, the potential is much greater than that. The great challenge of course remains, that high-resolution measurements are very sparse.

Because the MOSAiC mission provided us with an unmatched opportunity for training and testing algorithms with measured labels over a long time period, this study has made obvious that there is considerable room for improvement even with modern deep learning algorithms. It needs to be mentioned, that due to the spatial constraint to the area near the MOSAiC floe, the training dataset does not capture the full extent of possible winter ice conditions in the Arctic, thus we cannot expect the classifier to perform equally well on a pan-Arctic scale. Instances of OW/YI are very sparse and their entire span of possible conditions and consequent radar response is not covered well by data. Since a better in-situ dataset is probably not going to emerge in the near future, it is clear that measured labels alone are not enough to train a stable algorithm that can deal with the full span of ice conditions. It seems that to achieve this, one would need to leverage a great number of scenes without labels. Semi-supervised and self-supervised approaches come to mind. Some first examples of their development exist for optical data by HAN et al. (2019), ice and open water discrimination from SAR in Li et al. (2015); Khaleghian et al. (2021a) and for sea ice classes from SAR in Imber (2022).

## 5 Conclusions

The MOSAiC expedition enabled the generation of a large dataset (ca. 20 million data points) of SAR acquisitions and appropriate labels delineated from in-situ laser scanning measurements. It has become clear that both the freeboard and the above

snow surface roughness (at correlation lengths of 50 cm) are only weakly correlated with X-Band SAR backscatter, with average $R^2$ values of 0.124 and 0.043 respectively. We have shown that deep-learning segmentation approaches such as the Unet can approximate these labels from the SAR measurement at accuracies around 68%. We thus measured the performance of

330 modern network architectures on a representative set of labels for the first time and it is much more difficult to classify at this high detail than at coarser label resolution (e.g ice charts). From the performances of the different models, we can conclude that the semantic segmentation approaches advantage of being able to make use of the spatial relationship of predictions is crucial (20% accuracy) to the generalising to unseen regions. It is notable that these label distributions at the scale of the measurement resolution are not contained in ice charts or human annotations, which suggests that classifying accurately at the resolution of

335 the SAR measurement when trained on human-annotated labels is improbable. As a more comprehensive dataset than created here is unlikely to be acquired in the near future, newly developed classifiers aiming at classification at the resolution of the sensor will need to find some way to gain access to the spatial ice type distributions to be successful.

## Appendix A:  List of Helicopter Flights

| 1 | 20191020_01_PS122-1_2-167 |
|---|---|
| 2 | 20191119_01_PS122-1_8-23 |
| 3 | 20191130_01_PS122-1_9-98 |
| 4 | 20191224_01_PS122-2_17-98 |
| 5 | 20191225_01_PS122-2_17-99 |
| 6 | 20191228_01_PS122-2_17-101 |
| 7 | 20200107_01_PS122-2_19-44 |
| 8 | 20200108_01_PS122-2_19-46 |
| 9 | 20200108_03_PS122-2_19-52 |
| 10 | 20200116_01_PS122-2_20-52 |
| 11 | 20200121_01_PS122-2_21-41 |
| 12 | 20200123_02_PS122-2_21-78 |
| 13 | 20200128_01_PS122-2_22-16 |
| 14 | 20200204_01_PS122-2_23-14 |
| 15 | 20200212_01_PS122-2_24-31 |
| 16 | 20200217_02_PS122-2_25-8 |
| 17 | 20200227_01_PS122-3_29-49 |
| 18 | 20200318_01_PS122-3_32-42 |
| 19 | 20200408_01_PS122-3_35-49 |
| 20 | 20200423_01_PS122-3_37-63 |

**Table A1.** List of the 20 helicopter flights used in this research. Data is published in Hutter et al. (2022).

## Appendix B:  Network Architectures

We briefly present the network architectures used in this investigation. We make use of the following conventions to keep the figures concise. FCX is short for a fully connected layer with X neurons. ConvX x Y denotes a 2D convolutional layer with filter sizes X and number of filters Y. Unless otherwise specified the convolutional layers have stride 1. If a layer has multiple inputs, they are concatenated before being parsed to the layer.

| Input 256 |
|:---:|
| Conv3 x 64 |
| Conv3 x 64 |
| Maxpool2 |
| Conv3 x 96 |
| Conv3 x 96 |
| Maxpool2 |
| Conv3 x 128 |
| Conv3 x 128 |
| Conv3 x 128 |
| Maxpool2 |
| Conv3 x 192 |
| Conv3 x 192 |
| Conv3 x 192 |
| Maxpool2 |
| Conv3 x 256 |
| Conv3 x 256 |
| Conv1 x 256 |
| Maxpool2 |
| FC256 |
| FC128 |
| FC128 |
| SoftMax |

**Table B1.** VGG16 architecture as used in the paper. Published in Simonyan and Zisserman (2015). The ReLU activation is used throughout the network. The padding is set to 'same'.

| Input 256 | SoftMax |
|:---:|:---:|
| Conv3 x 32 | Conv3 x 32 |
| Conv3 x 32 | Conv3 x 32 |
| Maxpool2 | TConv2 x 32 |
| Conv3 x 32 | Conv3 x 32 |
| Conv3 x 32 | Conv3 x 32 |
| Maxpool2 | TConv2 x 32 |
| Conv3 x 48 | Conv3 x 48 |
| Conv3 x 48 | Conv3 x 48 |
| Maxpool2 | TConv2 x 48 |
| Conv3 x 64 | Conv3 x 64 |
| Conv3 x 64 | Conv3 x 64 |
| Maxpool2 | |
| Conv3 x 96 | TConv2 x 64 |
| Conv3 x 96 | |

**Table B2.** The Unet architecture as used in this paper and published in Ronneberger et al. (2015). The ReLU activation is used throughout the network and the padding is set to 'same' where applicable.

| |
|---|
| Input 256 |
| Conv4 (stride 4) x 96 |
| ConvNx x 96 |
| ConvNx x 96 |
| ConvNx x 96 |
| Conv2 (st 2) x 96 |
| ConvNx x 192 |
| ConvNx x 192 |
| ConvNx x 192 |
| Conv2 (st 2) x 192 |
| ConvNx x 384 |
| ConvNx x 384 |
| ConvNx x 384 |
| ConvNx x 384 |
| ConvNx x 384 |
| ConvNx x 384 |
| ConvNx x 384 |
| ConvNx x 384 |
| ConvNx x 384 |
| Conv2 (st 2) x 384 |
| ConvNx x 768 |
| ConvNx x 768 |
| ConvNx x 768 |
| GlobalAvgPool2D |
| LayerNorm |
| SoftMax |

where:

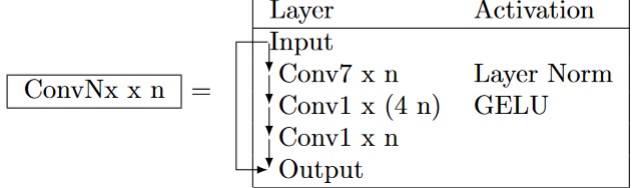

| Layer | Activation |
|---|---|
| Input | |
| Conv7 x n | Layer Norm |
| Conv1 x (4 n) | GELU |
| Conv1 x n | |
| Output | |

ConvNx x n =

**Table B3.** The ConvNext-T architecture used in this paper. Developed in Liu et al. (2022).

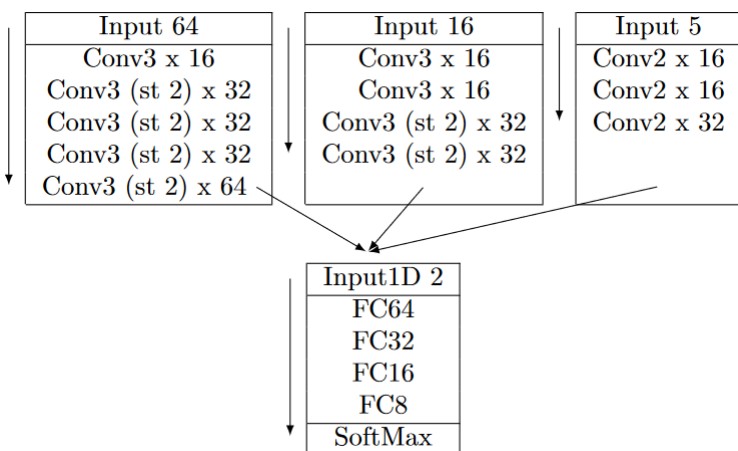

**Table B4.** The custom CNN architecture from Kortum et al. (2022) used in this paper. The inputs at different scales are flattened and concatenated before being output to the fully connected layers. Leaky ReLU is used for activation and padding is set to 'valid'. The 16x16 pixel input is downscaled from the original scene by factor 5 and the 64x64 pixel input is a square cutout that is rescaled so that the width of the entire scene is 64 pixels. The 1D input contains the relative coordinates of the pixel in the 64x64 pixel input.

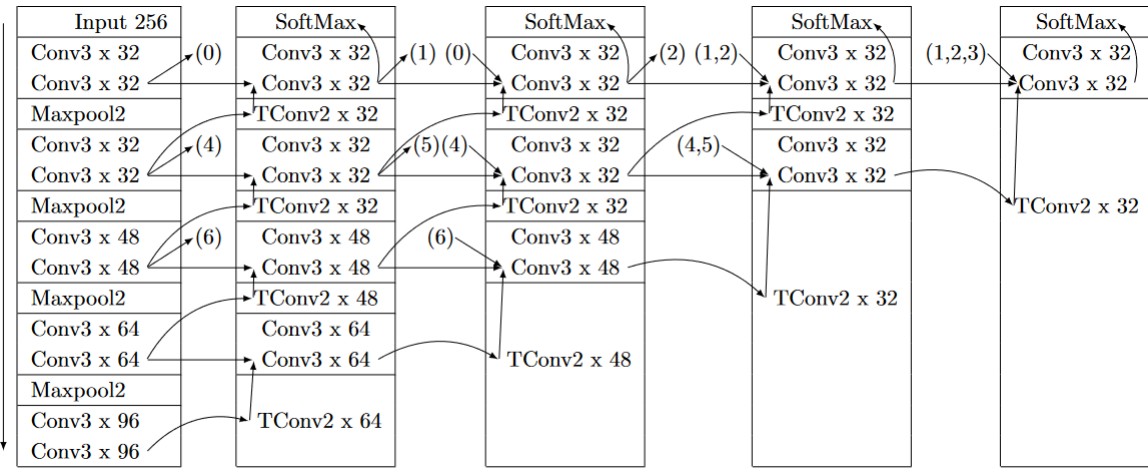

**Table B5.** Unet++ architecture used in this paper, published in Zhou et al. (2018, 2019). Note that the left column is identical to the downwards convolution side of the regular Unet and the lowest rows from left to right form the upwards side of the Unet. The Unet++ then uses extra layers in between to extend the architecture. All layers within a cell are considered to be a block, so they are all executed before parsing the output to the next block. All layers marked 'Softmax' are averaged before the final linear layer and the softmax are applied. ReLU is used as the activation function throughout and the padding is set to 'same'.

*Author contributions.* Karl Kortum - Conceptualization, Formal Analysis, Investigation, Methodology, Writing - original draft.

Suman Singha - Conceptualization, Data curation, Funding acquisition, Project administration, Supervision, Writing - review and editing.

Gunnar Spreen - Funding acqusition, Supervision, Writing - review and editing.

Nils Hutter - Data curation, Writing - review and editing.

Arttu Jutila - Data curation, Writing - review and editing.

Christian Haas - Funding acqusition, Supervision, Writing - review and editing.

*Competing interests.* Some authors are members of the editorial board of The Cryosphere. The peer-review process was guided by an independent editor, and the authors have also no other competing interests to declare.

*Acknowledgements.* This study was funded by Deutsche Forschungsgemeinschaft (DFG) under project name 'MOSAiCmicrowaveRS' (SI 2564/1-1 and SP 1128/8-1).

Data used in this manuscript was produced as part of the international Multidisciplinary drifting Observatory for the Study of the Arctic Climate (MOSAiC) with the tag MOSAiC20192020 and Project_ID: AWI_PS122_00..

We thank all persons involved in the expedition of the Research Vessel Polarstern during MOSAiC in 2019-2020 as listed in Nixdorf et al. (2021) Nixdorf et al. (2021).

TerraSAR-X images used in this study were acquired using the TerraSAR-X AO OCE3562_4 (PI: Suman Singha).

German Federal Ministry of Education and Research (BMBF) project IceSense—Remote Sensing of the Seasonal Evolution of Climate-relevant Sea Ice Properties (03F0866A).

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
