# Peer review of "SAR Deep Learning Sea Ice Retrieval Trained with Airborne Laser Scanner Measurements from the MOSAiC Expedition"

_The Cryosphere, 2023_

## Referee Comment (RC2)

Review of
'SAR deep learning sea ice retrieval trained with airborne laser scanner measurements from the MOSAiC
Expedition'

This paper looks at the retrievals of sea ice classes, corresponding to new ice/open water, level/first
year ice and deformed/multi-year ice using a set of X-band SAR images and airborne laser scanner data
from the MOSAiC expedition. This is a good goal, and relevant at this point in time. The dataset
is unique and the study is interesting, although in some places the details are not sufficient and the
message is not clear. For example, the introduction indicates the paper will address the longstanding
issue of sea ice classification, which has been hindered due to the lack of ground truth data and the
fact that the sea ice is drifting and deforming. The authors point to the deficiencies of the ice charts as
an example of inadequate labels. While there are many issues with ice charts, this feels a little discon-
nected because the chart labels are different than the ones used (i.e. the classes are different), and the
dataset used for the paper here is also quite specialized in comparison to what is available to make an
ice chart (which are typically C-band ScanSAR wide data, and other data sources). Regarding detail,
much more detail is needed about the networks chosen and how they were trained so we can assess
how systematic the comparison is, and if anything can be added to the discussion regarding unique as-
pects of the chosen architectures than can be better used, given that the test accuracies are not that high.

Major comments

- The SAR data (X-band stripmode SAR) is very niche (not everyday data). It is also high spatial
  resolution. Why was X-band used? What property does X-band have in comparison to C-band or
  L-band that make it suitable for this problem. I realize X-band was likely easier to obtain than
  other data, but how different would the results be if one were to use the more common C-band
  data at a coarser spatial resolution?

- line 92 'time between satellite measurements and helicopter measurements' how small is this? Is
  it really sufficient for precise for the features to overlap 'perfectly' across the entire swath for each
  image? For example on line 89 it says it is accurate to a couple of meters (is this +/- a couple
  of meters? So 4 meters?). Given that the SAR pixel spacing is 3.5 m, when classification is done
  pixel-wise, this may not be a wide margin (depending on what the accuracy means). If the authors
  think this is an issue it should be noted. Would some architectures be more sensitive to this than
  others?

- The assumption of a Gaussian distribution for freeboard and reflection does not reflect the PDFs
  shown in the paper. The PDFs shown in Figure 2 are really not Gaussian. A different type of
  distribution should be used or the impact of this assumption on the analysis should be assessed.

- I also don't follow how a Gaussian distribution allows one to infer uncertainties (unless the authors
  are referring to the use of 'soft' vs 'hard' labels, or probabilities rather than discrete classes). How
  does this help their overall objective (was there a comparison with using discrete labels?)

- Better justification of the choice of the architectures and implementation details are needed. While
  the Unet has been widely used for related sea ice studies, for ConvNext, VGG16, the CNN and the
  Unet variants it's not clear what special features these architectures have that make them suitable
  for the problem. For example, ConvNext is an improvement over ResNet, but in what way and how
  does this motivate using it here? In addition it's not clear if the models are trained from scratch
  or fine-tuned, or if the same stopping criteria are used for all of them? Could the authors show the
  training accuracies as well, for comparison with the test accuracies in Table 1.

- line 173 the kernel size of ConvNext is thought to be part of the problem with the poor model
  predictions. Could this not be changed and the hypothesis verified, or was there a reason for
  choosing ConvNext for which that kernel size or maintaining the overall architecture is important
  (for example did the authors consider it out of scope to modify established architectures)?

- lines 120-123 refer to the methods used to handle the class imbalance, but then on line 192 it
  says that the OW/YI classifications are the most error prone, 'expected because they are the most

sparse' in the dataset - does this imply that the methods used for imbalance were not effective, or could it be that the signatures of OW/YI were quite variable and not well represented given the sample size?

- line 154 The authors minimize the KL divergence instead of the cross entropy, which is more typical for a multi-class classification problem. I think the KL divergence should yield the same result as the cross-entropy (based on the definition of KL divergence), but if not then the could the authors explain their motivation.

- line 158 The validation set is very small (5% of data) - why is this so? Is it large enough to represent the general characteristics seen in the data (how was it chosen)?

- line 185 'generative models' - I don't think generative models were used in this study,

- the custom CNN architecture from Kortum et al. 2022 (shown in Appendix) uses a different size of input patch (smaller) than the other architectures. This leads to sharper, possibly noisier, predictions because the context over which the features are learned is smaller for the smaller patch (local or patch-size features are emphasized over global context). Similar results were found in Radhakrishnan et al. 2021 (reference below)

- line 69 'mow the lawn' pattern - if this is known terminology could a reference be given?

- Figure 1: Could something other than greyscale be used? The legend says white regions indicate no data, but it is hard to tell this apart from the very light grey. Also, there is no red circle.

- Figure 2: The PDFS shown are very smooth. Are fits over all the data points in the image shown to the left?

- Figure 4: This might be better placed as a subset of Fig 2.

- Figure 6: This is interesting in that the cyan regions are very irregular early in the season (November) and more linear later (in March) - is this typical?

K. Radhakrishnan, K. A. Scott and D. A. Clausi, "Sea Ice Concentration Estimation: Using Passive Microwave and SAR Data With a U-Net and Curriculum Learning," in IEEE Journal of Selected Topics in Applied Earth Observations and Remote Sensing, vol. 14, pp. 5339-5351, 2021, doi: 10.1109/JS-TARS.2021.3076109.

---

## Author Comment (AC1)

First of all, we would like to thank the reviewer for the expertise and time taken to review the manuscript. In general, we have considered all comments carefully and made changes to the manuscript to incorporate the improvements. In the following responses are given to comments where some additional detail is needed.

**Broad Comments:**

The manuscript covers an exciting topic that it is an open problem in the sea ice and SAR community. However, the text itself is currently lacking some much-needed details that would help the reader identify the full contributions of this work. The authors state in the Introduction that they are assessing existing deep learning approaches for sea ice classification in SAR imagery by testing them on a more reliable data set. The goal seems to be to advance our understanding of which deep learning approaches are actually advantageous for sea ice classification, leading to a standard for the community. However, there wasn't enough detail provided about the previous studies nor this study's methods to know if enough was kept consistent when repeating the analysis to have comparable results (e.g., did the previous studies also use X-band data? Were model parameters and frameworks kept consistent? etc.). The authors provide some discussion towards their differing results, especially in terms of resolution, but more information would be useful.

Establishing a new standard for the community is a little bit too lofty a goal for this study, as the best data driven solutions will always depend on the data, which will not be the same as in this study. The aim is to disseminate as much information as possible from the unique opportunity provided by the dataset created in this study to inform algorithm choices in the future and too offer a less idealised perspective of sea ice retrieval from SAR, than what is typically shown with human annotations used as ground truth – where labels are coarse and heavily influenced by human interpretation. In these scenarios typically regions are separated, because they look different on SAR, whilst in this study regions are separated because of different measured ice properties. As a result, the performance of the models in this study is more indicative of true dissemination of ice properties and not influenced by human labelling bias.

Concerning the consistency of methods: We have opted to use architectures that have been used for sea ice classification, but cannot go to the full length of effort to entirely reproduce many of the different variations that have been published on in the past. This would not be fruitful anyways, however, because of the difference in datasets that they were optimised for. The question we are trying to answer, and this will be posed more pointedly in the manuscripts next iteration, is: How do different CNN architectures perform on labels derived from measurements (not human interpretation) and how does that influence future algorithm choices if the aim is to produce classifications near the resolution of the sensor?

The answer to that question in the manuscript is, that real measurement derived classes are much more difficult to disseminate than human annotations – showing that human bias has had a large influence on studies of the past and the resulting models cannot be assumed to perform near the resolution/fidelity of the SAR sensor. Additionally, centre-pixel classifiers are significantly inferior at this task than semantic segmentation models, that can make use of intra-label dependencies.

**Specific Comments:**

**Section 1. Introduction**

1. Page 2, Figure 1: The caption mentions a red circle, but it was not visible to me. Perhaps check printed color (and consider using a different color for those who are colorblind).

This has been corrected.

2. Page 3, lines 47-48: Traditionally, these types of datasets tend to be too sparse to provide robust training sets for image-based ML models. How does this dataset differ?

Because of the efforts made during the MOSAiC expedition and the subsequent collocation for this work, the dataset used here is far larger than any previously used data derived from measurements. However, it still suffers from a loss of generality from being constrained to certain region. It is none-the-less most likely the most complete (as in large) collocated dataset that we will be able to synthesise at least until another expedition of the scope of MOSAiC comes along (which could be decades).

3. Page 3, line 57: How close is near-coincident? An example in parentheses would be helpful here.

This has been updated in the text and is 7 hours on average, with a range of 0-24 hours.

**Section 2. Methodology**

1. Page 3, first paragraph: More information is needed here on how the TSX data is processed (what corrections were performed, filtering, etc.) and what the effective resolution of the data is after such processing. Were you considering backscatter as your data? If so, what was it normalized to (e.g., , etc.)?

Thank you, this has been added to the next version of the manuscript.

2. Page 3, first paragraph: It is stated that only co-polarized channels are used. How do you mitigate effects from wind-roughened waters and other confounding factors which are more prevalent in co-polarized data?

The models have full access to both channels and contextual data to mitigate this as best as possible. Both configurations have advantages and disadvantages that a retrieval model will have to deal with.

3. Page 3, line 68: The ALS dataset seems to focus on the winter seasons. Is any comparison done for seasonality transfer?

Unfortunately, in summer the ship (used as coordinate system origin) moved significantly more during the warmer seasons and thus accurate geolocation here is more difficult. The same trained models would certainly not work in the warmer seasons, but I would expect the discrepancy between centre-pixel and semantic segmentation models to persist if both were retrained on summer data.

4. Page 3, second paragraph: What was the regional coverage of the ALS dataset? Did it cover all representative portions of the Arctic, or was it constrained to a particular region?

It was constrained to the region (~15km radius) around the central observatory (floe) of the MOSAiC Expedition (this is what enabled the colocation).

5. Page 3, second paragraph: A verbal comparison between ALS dataset footprint size versus the footprint of a TSX image would be useful for context.

This has been added to the data section (TSX 50x15km, ALS 10x5km).

6. Page 4, line 87: Was a distribution analysis of the freeboard and roughness per SAR pixel done to ensure that the mean and standard deviation are representative statistical measures for these data? If the distribution is heavily skewed, it may be more appropriate to use the median, for example.

The median and mean of the distribution, where on average within a percent of the span of the respective variables, so that a Gaussian description seems adequate. We have added some text in the manuscript mentioning this.

7. Page 4, line 92: Again, it would be helpful to have an example of these time differences (minutes, hours?).

It is typically on the order of hours. Within the dataset we did not see a drastic event that significantly impacted the ability to collocate the data. The relevant variables also should be largely unchanged within this time span.

8. Page 4, line 95: It would be good to explicitly name the conventions you are pulling from and provide an associated reference.

Conventions was the wrong word to be used here. We have changed this to 'names' it is just to allow an easier mental abstraction of the classes rather than to keep mentioning the thresholds used to distinguish them.

9. Page 5, line 111: It is unclear to me why you are assuming a Gaussian distribution when the density functions in Figure 2 are non-Gaussian. Your metrics (e.g., standard deviation) are likely to be heavily influenced by outliers.

The Gaussian distribution is only assumed locally to estimate the uncertainties in the derived classes. Whilst the global distributions are non-gaussian (as they are bounded), locally the gaussian description seemed adequate (see answer to comment 6 above) to achieve this. It is worth mentioning that the effect of this simplification on the loss function is minimal, so the model training is hardly affected by the small error we are introducing by simplifying the uncertainties to a standard deviation.

10. Page 6, Figure 3: Please add the mathematical notations for freeboard and SAR backscatter to the caption here so it is clear what the figure axes are referring to.
11. Page 6, Figure 4: Similarly, it would be good to note in the caption that PDF refers to probability density function for the general reader.
12. Page 7, line 123: Does favoring equal class performance affect how the model will perform operationally, where classes are almost always not equal?

Absolutely, the way it is set up here gives the most representative performance for unseen regions. If the aim was to make the most accurate classifications of the regions, the balancing would have to be approached differently.

13. Page 7, line 125: This sentence is unclear to me. Perhaps the second mention of "backscatter" should actually be "topography"?

Thank you, you are completely right. This has been corrected.

**Section 3. Results**

1. Page 8, first paragraph: Referring to the VGG16, etc. as pixel-wise classification approaches is confusing here, especially since under the Section 2.3 you describe these classification approaches as predicting over patches as a whole (or just the center), and the segmentation approaches as predicting a label for every pixel in a patch. Given that, seems backwards to refer to the center-pixel classification approaches as pixel-wise classification. I would try using a different descriptor if possible.

This is a very good idea and we have adopted it for the manuscript. Thank you!

2. Page 9, Figure 5: Do you have ground truth labels for this example? If so, it would be helpful to include them in the figure.

We will choose a different scene where ground truth is available. Thank you for pointing this out.

**Section 4. Discussion**

1. Page 10, line 204: Can you elaborate more on how your results compare to the results from previous studies that you are replicating?

As no specific study is being replicated, we need to be careful with our wording. We have added some extra discussion to establish the differences and conclusions to be drawn in comparison with existing studies relying on manual annotations.

2. Page 11, Figure 7: A reference for the misclassification of water and old ice being a common issue would be helpful to include here.

Yes, thank you for pointing this out, we have added a reference.

3. Page 12: It would be great to see some discussion on how the temporal span of the dataset may affect the results, as well as how you expect the results to change (or not change) when applied to different seasons. For example, do you think certain models would be more robust to the existence of melt ponds on older ice during the summer months?

Good idea! We have added some more discussion. To briefly recap I think that the core result of the usefulness of intra-label relationships for ice classification will be particularly relevant for surface types with characteristic shapes, such as melt ponds (in high resolution data) and leads. In general, the backscatter signatures themselves will be even less reliable in the summer months, so therefore I would expect the segmentation models to have an even bigger advantage there.

4. Page 12: Likewise, a discussion on the spatial coverage of the ALS dataset and how that could affect your results, especially when compared to previous studies, would be useful. Did this dataset only cover a particular region of the Arctic, and would you expect results to differ if you had data from other regions?

We have added some more discussion about this. The core takeaway is, that of course this dataset is not representative of the entire Arctic, but the distributions and shapes of different classes will have the same complexity in other regions. Thus, the core messages concerning the strong bias of manual labels translating to classifiers persists. The discrepancy between centre-pixel classification and

semantic segmentation models is also expected to be representative for the entire ice classification domain regardless of region.

Finally, we would like to once again thank the reviewer for a thorough and insightful review of the manuscript. The comments are very helpful and have/will significantly improve/d the manuscript from both a scientific and a didactic standpoint in our opinion. Sincerely, thank you!

---

## Author Comment (AC2)

First of all, I would like to thank the reviewer for taking the time and effort to read and review the manuscript and lending their expertise. Their comments are much appreciated.

This paper looks at the retrievals of sea ice classes, corresponding to new ice/open water, level/first year ice and deformed/multi-year ice using a set of X-band SAR images and airborne laser scanner data from the MOSAiC expedition. This is a good goal, and relevant at this point in time. The dataset is unique and the study is interesting, although in some places the details are not sufficient and the message is not clear. For example, the introduction indicates the paper will address the longstanding issue of sea ice classification, which has been hindered due to the lack of ground truth data and the fact that the sea ice is drifting and deforming. The authors point to the deficiencies of the ice charts as an example of inadequate labels. While there are many issues with ice charts, this feels a little disconnected because the chart labels are different than the ones used (i.e. the classes are different), and the dataset used for the paper here is also quite specialized in comparison to what is available to make an ice chart (which are typically C-band ScanSAR wide data, and other data sources). Regarding detail, much more detail is needed about the networks chosen and how they were trained so we can assess how systematic the comparison is, and if anything can be added to the discussion regarding unique aspects of the chosen architectures than can be better used, given that the test accuracies are not that high.

We have rewritten part of the introduction to clarify the aims of the paper. It does by no means solve the issues of sea ice classification: If anything, it offers a measurable disconnect between the commonly reported accuracies on manually or ice chart data and labels derive from measurement, which are not affected by human labelling biases. The connection to ice charts should be seen as a contrast as they are the only widely available source of label information. We take your comment with the amount of detail of the architectures very seriously. However, optimizing these architectures would offer little benefit in the long run, as the classifiers trained here will never be performant on a larger scope of ice conditions due to the constraint to a region near the MOSAiC floe and further optimization is very much dependent on the underlying data. Thus, statements about the usefulness of optimization techniques cannot be assumed to generalise.

**Major comments**
• The SAR data (X-band stripmode SAR) is very niche (not everyday data). It is also high spatial resolution. Why was X-band used? What property does X-band have in comparison to C-band or L-band that make it suitable for this problem. I realize X-band was likely easier to obtain than other data, but how different would the results be if one were to use the more common C-band data at a coarser spatial resolution?

As the you rightly suggested, other available data would not have been available at the resolution or frequency to enable high spatial overlap (in terms of pixels) at small enough time differences. At higher wavelengths, especially L-Band we might expect at least a higher correlation between radar backscatter and surface roughness, as this was measured at spatial intervals of 0.5 metres, this would probably translate to higher classification accuracy for deformed ice. The complexity of the spatial distribution of classes we would not expect to drastically change at these coarser resolutions. This can be argued with the idea of fractality of structures of the Arctic sea ice across these length scales (although at resolutions as small as 3.5 m this might start to break down). Thus, the core advantage of being able to use intra-label relationships, that semantic segmentation models have over centre-pixel classifiers, is probably similarly useful even at larger length scales. There is also little evidence to suggest that the classification problem becomes easier at larger scales.

• line 92 'time between satellite measurements and helicopter measurements' how small is this? Is it really sufficient for precise for the features to overlap 'perfectly' across the entire swath for each image? For example on line 89 it says it is accurate to a couple of meters (is this +/- a couple of meters? So 4 meters?). Given that the SAR pixel spacing is 3.5 m, when classification is done pixel-wise, this may not be a wide margin (depending on what the accuracy means). If the authors think this is an issue it should be noted. Would some architectures be more sensitive to this than

others?

From a visual perspective, the features really do line up perfectly between the two measurements over the entire helicopter data. Any residual inaccuracies are on the scales of a single pixel and as most features are significantly larger than that the influence of those inaccuracies should be small. Great comment on relating this to the architectures. The central pixel classifiers are probably less affected, as they have no intrinsic knowledge of classifying exactly the centre. Realistically they have a smaller 'effective resolution' – not necessarily classifying what is in the central pixels but rather what is in a wider notion of the centre of the image, therefore small shifts in the matching would have less of an effect on training.

• The assumption of a Gaussian distribution for freeboard and reflection does not reflect the PDFs shown in the paper. The PDFs shown in Figure 2 are really not Gaussian. A different type of distribution should be used or the impact of this assumption on the analysis should be assessed.

Note that the gaussian assumption is only used for local distributions (at a single TS-X Pixel), which are usually quite far from the bounds of the global pdfs. Some analysis has been added to the manuscript, that showed the mean and medians of these distributions to be different by less than a percent of the span of the distribution on average. This leads us to believe the impact of simplifying to gaussian distributions is minimal.

• I also don't follow how a Gaussian distribution allows one to infer uncertainties (unless the authors are referring to the use of 'soft' vs 'hard' labels, or probabilities rather than discrete classes). How does this help their overall objective (was there a comparison with using discrete labels?)

You are correct in the assumption of using 'soft' labels. Originally, we based this on past experiments. To follow up on your comment, we conducted a comparison with discrete labels which showed reduced accuracies by about eight percent for the best performing U-net architecture, which is quite significant. This confirmed our past experiences and will be added to the paper.

• Better justification of the choice of the architectures and implementation details are needed. While the Unet has been widely used for related sea ice studies, for ConvNext, VGG16, the CNN and the Unet variants it's not clear what special features these architectures have that make them suitable for the problem. For example, ConvNext is an improvement over ResNet, but in what way and how does this motivate using it here? In addition it's not clear if the models are trained from scratch or fine-tuned, or if the same stopping criteria are used for all of them? Could the authors show the training accuracies as well, for comparison with the test accuracies in Table 1.

VGG16 style architectures have also been used for ice charting in the past. The ConvNext model is a natural progression for such architectures, the same way the Unet++ is for the Unet, thus including them was found to be sensible. Although they are quite varied it strengthens the case of different results from centre-pixel vs semantic segmentation models that is highlighted in this research. The models were all trained from scratch and stopped under the same conditions. Due to premature stopping of the training run based on validation set results, training set accuracies were only a little better than on the test set (0-5%). Thank you for the comment, we will make sure to have more detail about the training procedure and the implementations in the next iteration of the manuscript.

• line 173 the kernel size of ConvNext is thought to be part of the problem with the poor model predictions. Could this not be changed and the hypothesis verified, or was there a reason for choosing ConvNext for which that kernel size or maintaining the overall architecture is important (for example did the authors consider it out of scope to modify established architectures)?

In general, the aim was not to modify the existing architectures, as a lot of time could be spent on optimisation, that would in the end not be meaningful for further research, as they would depend quite strongly on this unique dataset. It would also open up the question how much the optimisation contributed to the result and how much effort would be spent on optimising etc. However, out of

interest, we have redone the calculations with smaller kernel sizes: ConvNext with 3x3 kernels performed better by around 5% accuracy, whilst the population variance remained similar.

• lines 120-123 refer to the methods used to handle the class imbalance, but then on line 192 it says that the OW/YI classifications are the most error prone, 'expected because they are the most sparse' in the dataset - does this imply that the methods used for imbalance were not effective, or could it be that the signatures of OW/YI were quite variable and not well represented given the sample size?

Although the balancing strategies help mitigate the problem, they cannot fully overcome such a heavy imbalance. The variable signatures are definitely also a problem, that should be mentioned, thank you!

• line 154 The authors minimize the KL divergence instead of the cross entropy, which is more typical for a multi-class classification problem. I think the KL divergence should yield the same result as the cross-entropy (based on the definition of KL divergence), but if not then the could the authors explain their motivation.

You are right the optimization will be the same. The parameter itself gives more accurate representation of the amount of information that is lost, rather than just the accuracy of the result. We will rephrase some sentences to better reflect this.

• line 158 The validation set is very small (5% of data) - why is this so? Is it large enough to represent the general characteristics seen in the data (how was it chosen)?

In general, the amount of data we have available is not large and we wanted to make use of it as efficiently as possible. The sole purpose of the validation set is to set a 'fair' stopping point for all models, which is why we kept it the smallest of all the dataset. It is a random slice of the data, so it was not chosen in a deliberate manner. This also ensures less human influence on the results.

• line 185 'generative models' - I don't think generative models were used in this study,

Apologies for the wording, you are right it is misused here.

• the custom CNN architecture from Kortum et al. 2022 (shown in Appendix) uses a different size of input patch (smaller) than the other architectures. This leads to sharper, possibly noisier, predictions because the context over which the features are learned is smaller for the smaller patch (local or patch-size features are emphasized over global context). Similar results were found in Radhakrishnan et al. 2021 (reference below)

Thank you for the reference, we will follow up and incorporate this into the text. The context (in terms of distance) is not smaller because the inputs are down sampled at different rates. But the amount of contextual pixels that information could be drawn from certainly are. This architecture was quite heavily optimised for different ground truth data, which did not translate to the data used here.

• line 69 'mow the lawn' pattern - if this is known terminology could a reference be given?

It is the best descriptor we could come up with, the data product itself could be referenced but this would not help didactically. We can change to a more descriptive approach detailing how the data's narrow parallel swaths were acquired in such a manner that the individual subswaths had small overlap along the long side, until the desired area was completely covered.

• Figure 1: Could something other than greyscale be used? The legend says white regions indicate no data, but it is hard to tell this apart from the very light grey. Also, there is no red circle.

Thank you, we will improve this for better visual readability.

• Figure 2: The PDFS shown are very smooth. Are fits over all the data points in the image shown to the left?

Some Gaussian kernel smoothing has been applied here to smooth over the PDFs. We will update the text to reflect this

• Figure 4: This might be better placed as a subset of Fig 2.

Good suggestion, we will change this.

• Figure 6: This is interesting in that the cyan regions are very irregular early in the season (November) and more linear later (in March) - is this typical?

Interesting observation. The shape of leads is in some way's indicative of the breaking processes and rheology. We have thought about creating a lead-shape dataset to assess this quantitatively with some colleagues, but need to work on an efficient outlier detection scheme to automate this process.

K. Radhakrishnan, K. A. Scott and D. A. Clausi, "Sea Ice Concentration Estimation: Using Passive Microwave and SAR Data With a U-Net and Curriculum Learning," in IEEE Journal of Selected Topics in Applied Earth Observations and Remote Sensing, vol. 14, pp. 5339-5351, 2021, doi: 10.1109/JSTARS.2021.3076109.

Thank you very much for your helpful comments and interesting ideas. These are sure to improve the quality of the work and give some valuable ideas for future work. Your time and effort is very much appreciated!

---

## Author Response (AR1)

**Manuscript changes**

The changes are tied closely tied to the comments from the two reviews and are already outlined in the responses given there. For a brief summary, the main changes are listed below.

- More pointed formulation of goals in the outset of the study.
- Extended description of X-Band data and influence on results.
- Description of core features of different architectures.
- More detailed explanation of training procedure including new figure (figure 4). (We found a small bug in the data generators in the analysis, so we retrained the models which slightly changed the results).
- Inclusion of training data performances in results table.
- Added discussion of training data selection on findings.

For a detailed view of all changes see the latexdiff document '*changes.pdf'.

---

## Referee Report (RR1)

Second review of
'SAR deep learning sea ice retrieval trained with airborne laser scanner measurements from the MOSAiC Expedition'

The manuscript is significantly improved. I do have a few more dcomments. In addition, please read the entire manuscript carefully. I found a number of typos (some are listed below) but there may be others, and there are also spacing issues (words inserted without spaces).

- line 189 'of with' should be with

- line 191 'fully layers' should be fully connected layers

- line 195 'convolutions' should be convolution

- line 202 lower case on 'convolutional' (it is not a proper name like Gaussian)

- line 203 'inter-label' - this is different than in the reply to reviewers, where 'intra-label' was used. 'Inter' and 'Intra' label are quite different. Please clarify.

- line 278 'unknow' should be unknown

- The categorical cross-entropy vs KL divergence part is unclear. Looking at Table 1, the train/test accuracies and train/test KL divergence are quite different, and even for some models (e.g. ConvNext) the trends are different (in that training accuracy is higher than test for train while for KLD test accuracy is higher than train). Why is this the case? Then, when I look at the caption for the table it says 'standard deviation' has been shortened to 'std' - but there is no std in the table. Again on line 250 reference is made to the mean and spread of the KLD but on line 213 it is stated the the cross-entropy is minimized in training. Please clarify these points.

- The wording for how the labels are generated I still find confusing. On line 136 it is stated a Gaussian distribution is assumed for the freeboard and reflection measurements. But in figure 2 freeboard and surface roughness are shown. Do the authors mean a Gaussian distribution is assumed for freeboard and surface roughness measurements? Continuing through this paragraph where it says the Gaussian distribution is integrated above and below the threshold - I am not sure the reader knows what this threshold is. You might mean (for example in the top right of fig2) where the color changes at a freeboard elevation of 0.4 so you are integrating the distribution to this threshold to determine the soft label for FYI etc. However, there are two PDFs shown on the right of Fig 2 (one for freeboard and one for elevation) - so presumably you integrate both to get the three classes considered (OW/YI, FYI and SYI shown in Fig 4). But this is not clear from the text. In addition the threshold mentioned in the text (searching for threshold) is for backscatter. This could be clarified by revising the text leading up to figure 2 and adding some subfigure labels (a,b,c etc). There are two sets of PDFs in the figure (one for the freeboard and surface roughness, and the other for the backscatter). Using subfigure labels would help by stating 'integrating the PDF (fig 2a etc)' so the reader knows what PDF and what threshold you are talking about.

- In Fig 4 SYI is a class but in elsewhere it is MYI/DI.

---

## Author Response (AR2)

Thank you for your efforts of going through the manuscript and the further suggestions. The language notes have been adopted and the larger comments are answered below. I believe the changes made and extra figure added help the clarity of the manuscript.

1. line 203 'inter-label' - this is different than in the reply to reviewers, where 'intra-label' was used. 'Inter' and 'Intra' label are quite different. Please clarify.

Inter is correct, as we mean the relationships between individual (pixel) labels.

2. The categorical cross-entropy vs KL divergence part is unclear. Looking at Table 1, the train/test accuracies and train/test KL divergence are quite different, and even for some models (e.g. Con vNext) the trends are different (in that training accuracy is higher than test for train while for KLD test accuracy is higher than train). Why is this the case? Then, when I look at the caption for the table it says 'standard deviation' has been shortened to 'std' - but there is no std in the table. Again on line 250 reference is made to the mean and spread of the KLD but on line 213 it is stated the the cross-entropy is minimized in training. Please clarify these points.

The Kullback-Leibler Divergence (KLD) is a measure for the distance of the distributions – considering not only the class with the highest predicted likelihood, but also describing how close the predicted label probabilities are to the target probabilities (derived from measurement). In this case the lower the difference is, the closer the models have fitted to the measured distribution. Thus, the trends are very similar to the accuracy trends, wherein models with higher disparities between train and test accuracies also have a larger difference between training and test set KLDs and the test KLDs are always larger (worse) than the train KLDs. We have added some text and rewritten the caption to clarify this

3. The wording for how the labels are generated I still find confusing. On line 136 it is stated a Gaussian distribution is assumed for the freeboard and reflection measurements. But in figure 2 freeboard and surface roughness are shown. Do the authors mean a Gaussian distribution is assumed for freeboard and surface roughness measurements? Continuing through this paragraph where it says the Gaussian distribution is integrated above and below the threshold - I am not sure the reader knows what this threshold is. You might mean (for example in the top right of fig2) where the color changes at a freeboard elevation of 0.4 so you are integrating the distribution to this threshold to determine the soft label for FYI etc. However, there are two PDFs shown on the right of Fig 2 (one for freeboard and one for elevation) - so presumably you integrate both to get the three classes considered (OW/YI, FYI and SYI shown in Fig 4). But this is not clear from the text. In addition the threshold mentioned in the text (searching for threshold) is for backscatter. This could be clarified by revising the text leading up to figure 2 and adding some subfigure labels (a,b,c etc). There are two sets of PDFs in the figure (one for the freeboard and surface roughness, and the other for the backscatter). Using subfigure labels would help by stating 'integrating the PDF (fig 2a etc)' so the reader knows what PDF and what threshold you are talking about.

Some detail has been added to the manuscript and a figure has been added to clarify the situation. The general procedure is to take the globally defined thresholds which define classes and then check locally – for each SAR pixel – how likely it is that the ice there lies above or below these thresholds, by assuming that all measurements of the ALS sensor which are mapped to that pixel admit an

approximately Gaussian distribution. The refined text and figure I have included here for convenience:

"As detailed above, ice types are identified by thresholds in the reflectance, surface roughness or freeboard. The thresholds for the roughness and freeboard are indicated in the histograms in Fig. 4 by the different background colors. We can infer the probabilities of lying above or below a threshold for every pixel by assuming a Gaussian distribution of ALS freeboard and reflection measurements at each SAR pixel. From the 49 ALS measurements mapped to one SAR pixel, we compute the mean and standard deviation of the freeboard and can then compute the probabilities of lying below or above the globally defined freeboard thresholds by using the error function. Explicitly, we integrate the area under the curve of the estimated Gaussian probability density function, above and below the threshold. An example is shown in Fig. 3. Thus, we obtain 'soft labels' which give the probabilities of belonging to a certain class, rather than discrete classes. Assuming a Gaussian distribution allows us to also infer uncertainties of the surface roughness. One could have classified each of the 49 ALS measurements mapped to one SAR pixel and then used the relative occurrences as probabilities. However, this simplification to a Gaussian distribution leads to an inaccuracy of the probabilities (derived from freeboard) of only ~ 0.16% on average, but significantly increased computational efficiency."

[Figure]

**Fig:** Soft labels are derived for one SAR pixel by assuming a Gaussian distribution (colored) of the 49 ALS observations (grey histogram) inside of it and then integrating the area under the pdf curve above and below the threshold. In the given example the probabilities are close to 50%.